# Mapping Soil and Pasture Attributes for Buffalo Management through Remote Sensing and Geostatistics in Amazon Biome

**DOI:** 10.3390/ani12182374

**Published:** 2022-09-12

**Authors:** Gislayne Farias Valente, Gabriel Araújo e Silva Ferraz, Lucas Santos Santana, Patrícia Ferreira Ponciano Ferraz, Daiane de Cinque Mariano, Crissogno Mesquita dos Santos, Ricardo Shigueru Okumura, Stefano Simonini, Matteo Barbari, Giuseppe Rossi

**Affiliations:** 1Agricultural Engineering Department, Federal University of Lavras, P.O. Box 3037, Lavras 37200-900, MG, Brazil; 2Department of Agronomy, Federal Rural University of the Amazon—UFRA, P.O. Box 3017, Parauapebas 68515-000, PA, Brazil; 3Department of Soil Science, Rural Federal University of Pernambuco, Recife 52171-900, PE, Brazil; 4Department of Agriculture, Food, Environment and Forestry (DAGRI), University of Florence, Via San Bonaventura 13, 50145 Florence, Italy

**Keywords:** spatial variability, buffalo farming, precision livestock

## Abstract

**Simple Summary:**

Buffalo breeding in the Amazon biome can contribute significantly to local community development and, thus, is considered an essential income source. However, in Amazon regions, the inadequate breeding of these animals can lead to considerable negative consequences for the environment. Therefore, it is crucial to develop methodologies to improve animal management and grass yield. One of these methodologies is related to Precision Agriculture (PA), adapted for pasture and animal monitoring. Along these lines, we seek to utilize geostatistical techniques and remote sensing applications to better understand Buffalo grazing under a rotating system. In particular, we analyze forage Dry and Green Matter, as well as pH in pasture soils, demonstrating the obstacles against and advantages of the implementation of precise techniques for decision making and increasing grass productivity. We describe ways in which geostatistical soil pH mapping can be conducted, as well as the premises necessary to include remote sensing data in the analysis of pasture variables. Implementing these results in buffalo management systems can contribute to greater productivity and increasingly sustainable livestock.

**Abstract:**

The mapping of pastures can serve to increase productivity and reduce deforestation, especially in Amazon Biome regions. Therefore, in this study, we aimed to explore precision agriculture technologies for assessing the spatial variations of soil pH and biomass indicators (i.e., Dry Matter, DM; and Green Matter, GM). An experiment was conducted in an area cultivated with *Panicum maximum* (Jacq.) cv. Mombaça in a rotational grazing system for dairy buffaloes in the eastern Amazon. Biomass and soil samples were collected in a 10 m × 10 m grid, with a total of 196 georeferenced points. The data were analyzed by semivariogram and then mapped by Kriging interpolation. In addition, a variability analysis was performed, applying both the Normalized Difference Vegetation Index (NDVI) and Normalized Difference Water Index (NDWI) derived from satellite remote sensing data. The Kriging mapping between DM and pH at 0.30 m depth demonstrated the best correlation. The vegetative index mapping showed that the NDVI presented a better performance in pastures with DM production above 5.42 ton/ha^−1^. In contrast, DM and GM showed low correlations with the NDWI. The possibility of applying a variable rate within the paddocks was evidenced through geostatistical mapping of soil pH. With this study, we contribute to understanding the necessary premises for utilizing remote sensing data for pasture variable analysis.

## 1. Introduction

The Buffalo population in Brazil has been estimated to be 1.43 million animals, with a higher concentration in the northern region (546,777 animals). Pará state leads the production ranking in the north, being responsible for 39% of the national production of dairy and beef animals [1].

Buffalo breeding is an effective economic activity in Brazil, due to the adaptation of the species to several environments. In addition, buffalo farming has gained popularity due to the peculiar physico-chemical characteristics of their milk and meat [2,3,4]. They are commonly bred in extensive ecosystems consisting of native or cultivated pastures [5]

Pastures are forage resources, considered the main nutrient source in ruminant feeding and serving as a subject when aiming to maximize productivity [6,7]. The most-used forage grass in Northern Brazil pastures is *Panicum maximum* (Jacq.) cv. Mombaça, which stands out for its ability to produce dry matter and leaves [8,9,10]. Leaf production corresponds to the Green Matter (GM) component, and expressive development of this characteristic is desirable in forage production. Leaf production represents the botanical attribute of greater digestibility, and has a direct influence on animal performance [11]. The amount of GM is positively correlated with forage consumption and can be used to characterize pasture development [12,13,14]. Another important parameter for analyzing pasture development is dry matter (DM), derived as the forage mass present instantly above ground level per unit area without considering water content [15].

Grazing practices alter the forage structure and forage mass content [16]. Forage mass plays a crucial role in understanding feeding patterns. Therefore, above-ground forage mass parameters are often used as indicators of available forage quality and quantity for the animal unit, providing information for farm management [17]. Traditional forage mass analysis entails laborious and destructive actions [18,19]. These methods have low sampling precision, as they are based on random sample means without considering the spatial variation of biomass [20,21].

Consequently, farmers seek new technologies, better methods, and data to understand the impact of pasture management practices on optimizing forage yield [22]. Non-destructive, indirect methods have been systematically explored to measure grass yield and variability in the production field [23,24]. Among these methods, remote sensing technologies can provide vital information for forage mass estimation and pasture management [25].

Imaging by remote sensors provides spatial, spectral, and temporal resolutions that can facilitate new monitoring modes [26,27]. Combined with remote sensing information, grass biomass and variability measurements within the production field through precision agriculture technologies have been continuously explored [28,29]. These technologies can provide short-term estimates of culture biophysical characteristics [30,31]. The use of spectral information obtained from orbital images to detect changes in canopy structure is a well-established technology [32].

Analyses based on geostatistical techniques have also been used to determine spatial variability in the field [33], and are considered adequate for analyzing the spatial variability in the physical and chemical properties of soil and plants [34,35,36].

The combination of remote sensing technologies and geostatistical techniques can improve pasture system management. The spatial and temporal monitoring of pasture biomass production by non-destructive methods allows for seasonal production curve visualization, quality, and estimation of production level [37,38]. This information can determine the feasibility of new techniques, allowing for optimization of animal management practices in specific pastures for forage grass, such as those in the Amazon biome.

Field data from dry matter integrated with the data obtained by orbital sensors through vegetation indices can possess an expressive degree of importance [22,39]. These indirect biomass measurements provide information for pasture planning [40]. However, the methods used to estimate the forage mass and sample number influence the variability in the results [20]. Thus, it is necessary to consolidate an analysis methodology possessing accuracy and precision that is more assertive than the method used for forage mass evaluation.

This research was carried out to evaluate the application of techniques for obtaining the spatial variation of Mombaça grass biomass utilizing remote sensing technologies and geostatistical analyses. NDVI and NDWI vegetation indices and geostatistical methods were explored to map the variation in biomass yield of *Panicum maximum* (Jacq.) cv. Mombaça in a pasture under a rotational grazing system with buffaloes in the eastern Amazon.

## 2. Materials and Methods

### 2.1. Area of Study

The experiment was carried out at Açaizal farm, located in Parauapebas municipality, Pará, Brazil, following Universal Transverse Mercator (UTM) coordinates 623,667.46 E/9,311,972.51 N, zone 22 S (Figure 1). According to the Köppen classification [41], the regional climate is tropical monsoon climate (AW); that is, a rainy tropical region with rains concentrated in the summer and a dry season in the winter.

The experimental area of size 17,150 m^2^ was subdivided into seven parts (each of 2450 m^2^). In these paddocks, soil samples were collected, in order to characterize forage aerial parts and pH at different depths. (Figure 1). The soil was classified as Red–Yellow Argisol with a sandy loam texture [42] (Table 1).

The area consisted of a dairy buffalo production system under pasture cultivated with *Panicum maximum* (Jacq.) cv. Mombaça. They were managed using a rotational stocking grazing method, with an average stocking rate of 3 AU ha^−1^ (Animal Unit = animal of 450 kg live weight). This grazing method subdivides a pasture area into paddocks, which are subjected to controlled grazing periods (occupation) and rest periods (fallow) [43]. Each paddock in the experimental area was occupied by 24 h of grazing and 21 days of rest.

### 2.2. Data Acquisition

The spatial characterization of Dry Matter (DM), Green Matter (GM), and soil pH was performed by collecting data that were georeferenced using a GPS receiver. Sampling was performed in each paddock, excluding 2.5 m from the fence border and 5 m from the side border. In this way, a regular mesh spaced equidistantly at 10 m and with a sampling density of 28 points in each paddock was generated, for a total of 196 points (Figure 1).

#### 2.2.1. Soil pH

Soil samples were collected at 0–0.2 m, 0.2–0.3 m, and 0.3–0.4 m at the demonstrated georeferenced points (Figure 1). Subsequently, they were sent to a soil analysis laboratory to obtain the pH values in Potassium chloride (KCl), following the methodology proposed by Teixeira et al. [44].

#### 2.2.2. Forage Matter

Forage mass representative samples were obtained by cutting the forage according to animal rotation, performed at the end of the rest period in each paddock and one day before animal entry.

Samples were obtained to characterize the Dry Mass (DM) and Green Mass (GM) at the georeferenced points (Figure 1). The samples were obtained by cutting the forage, using wooden frames of 0.25 m^2^ systematically placed over sampling grid intersection points. The forage mass was placed in identified and weighed plastic bags, thus obtaining the total weight of fresh grass mass.

Green Matter (GM), expressed in ton/ha^−1^, was obtained according to the methodology adapted from Veras et al. [45]. The GM determination consisted of collecting subsamples of representative fresh grass mass at each sampling point. Then, for Dry Matter (DM) determination, the GM sample was submitted to leaf blade morphological separation, providing stem (stem + leaf) and dead material fractions. It was conditioned in a paper bag, identified, and put into an oven with air circulation at a temperature of 65 °C until a constant weight was reached, as described by Reis et al. [46]. After drying, the samples were weighed and converted into DM grass availability (ton/ha^−1^).

### 2.3. Geostatistical Analysis

Field data collected were subjected to geostatistical analysis, in order to characterize the spatial variation of the pH, DM, and GM variables, using the R Development Core Team software [47]. Subsequently, these data were submitted to semivariance analysis, in order to verify the existence of spatial dependence [48], as well as data interpolation by ordinary Kriging to create spatial variability maps [49]. The semivariance calculation was performed using Equation (1), as described by Matheron [50]:(1)$$\hat{\gamma }(h)=\frac{1}{2N(h)}\overset{N(h)}{\underset{i=1}{\sum }}{\left[Z\left(X_{i}\right)-Z\left(X_{i}+h\right)\right]}^{2}$$
where
$\hat{\gamma }(h)$ is the estimated semivariance at a distance *h*;*N*(*h*) is the number of experimental data pairs separated by a distance *h*;*Z*(*X_i_*) is the value determined at sample point *i*;*Z*(*X_i_* + *h*) is the value measured at point *i* plus a distance *h*.

The parameters of nugget effect (C_0_), sill variance (C_0_ + C_1_), and range (a) were obtained using the semivariogram equation, fitted according to graph behavior testing of linear, spherical, exponential, and Gaussian models. The choice of model was based on the highest coefficient of determination (R^2^) and lowest residual sum of squares (SQR) [51].

Data estimation confidence was assessed in terms of the Degree of Spatial Dependence (DSD), determined by participation of spatial variance in the total sample variance [52]. To assess the DSD, Cambardella et al. [53] developed a classification characterized by the ratio between the nugget effect (C_0_) and the sill variance (C_0_ + C_1_). The confidence can be considered weak with a DSD greater than 75%, moderate between 25% and 75%, and strong for DSD less than 25% [53].

### 2.4. Kriging Maps

Semivariogram adjustment was used to identify the spatial variability of DM, GM, and soil pH. These adjustments enabled the interpolation of data through ordinary Kriging and map construction. In this step, soil pH maps were constructed at depths of 0–0.20 m, 0.2–0.30 m, and 0.3–0.40 m, as well as maps of Dry Mass (DM) and Green Mass (GM).

The maps were constructed using the QGis software. The sampled points were interpolated to obtain an estimate (z*) consisting of a linear combination of neighboring measurement values (X0), represented by Equation (2):(2)$${z^{*}(x}_{0})=\overset{N}{\underset{i=1}{\sum }}\lambda_{i}z\left(x_{i}\right),$$
where

z* is the estimate;

x_0_ is a linear combination of neighboring measurement values;

N is the number of measured values involved in the z estimate (x_i_);

i is the weight associated with each measured value.

Kriging maps were prepared to compare spatial variation of pH at different soil depths, DM, and GM.

### 2.5. Obtaining and Processing Orbital Data

Analyses using data from orbital sensors were conducted after selecting a set of multi-spectral images from the SENTINEL 2B sensor, made available by the United States Geological Survey (USGS). Data from this sensor were used due to it being freely available and considering its resolution characteristics (i.e., spatial, spectral, and temporal). Images from SENTINEL 2B have a spatial resolution of 10 m in the spectral bands, as shown in Table 2. The set of images obtained were for 12/02/2017.

Initially, the images were selected with respect to image quality criteria (clouds and noise). Then, they were submitted to processing based on the Geographic Information System (GIS) QGIS software version 3.6. Raw image data processing comprised the first steps of obtaining multi-spectral data, conducting atmospheric corrections, and geolocation.

The calculations applied to obtain vegetative indices are influenced by the effects of reflectance scattering from the atmosphere; thus, in these cases, atmospheric correction should be carried out [54]. A modification (Dark Object Subtraction; DOS) was applied using a plugin in the QGIS 3.6 software, in order to correct this interference. After processing, the images were cropped to the area of interest for the calculation of vegetative indices.

The vegetative indices applied were the NDVI (Normalized Difference Vegetation Index) proposed by Rouse et al. [55] (Equation (3)) and the NDWI (Normalized Difference Water Index) presented by Gao [56] (Equation (4)):(3)$$\mathrm{NDVI}:\frac{\rho \mathrm{NIR}-\rho \mathrm{RED}}{\rho \mathrm{NIR}+\rho \mathrm{RED}}$$
whereNDVI is the Normalized Difference Vegetation Index;ρNIR denotes the Near-Infrared Band;ρRED denotes the Red Band.

(4)$$\mathrm{NDWI}:\frac{\rho \mathrm{NIR}-\rho \mathrm{SWIR}1}{\rho \mathrm{NIR}+\rho \mathrm{SWIR}1}$$
where
NDWI is the Normalized Difference Water Index;ρNIR denotes the Near-Infrared Band;ρSWIR1 denotes the Shortwave Infrared Band.

Vegetative indices are often used to describe plant development, vigor, and biomass [57]. Thus, the vegetative indices were applied to characterize the biomass in each grazing paddock before cutting. In addition, the correlations between the vegetative indices and matter variables (i.e., DM and GM) was assessed.

### 2.6. Regression Analysis

The pH, DM, GM, NDVI, and NDWI data were subjected to regression analysis, in order to evaluate the variables as a function of the other variables. In particular, regression analyses were performed between soil pH data at each depth (0–0.20 m, 0–0.30 m and 0–0.40 m) with DM and GM values, as well as between NDVI and NDWI with the forage variables DM and GM.

## 3. Results

### 3.1. Geostatistical Parameters

Geostatistical analyses make it possible to verify several characteristics of a data set [58]. Table 3 presents semivariogram adjustment parameters for soil pH in the soil layers 0–0.2 m, 0.2–0.3 m, and 0.3–0.4 m. All the fitted models showed strong DSD (DSD < 25%) [53], indicating the reliability of the semivariograms in explaining the variations in experimental data [59].

The pH at a depth of 0–0.2 m showed the smallest range (83.9 m), and fit the spherical model (Table 3). At depths of 0.2–0.3 m and 0.3–0.4 m (Table 3), the practical ranges were 256.91 and 298 m, respectively, better fitting the Gaussian model. Table 3 indicates the increase in range as the soil layer deepened.

The GM value adjustments were obtained according to the model and parameters characteristics presented in Table 4. The spherical model best fit the data set, having the smallest residual sum of square (RSS) and the highest coefficient of determination (R^2^).

The existence of spatial dependence for DM data can be observed in Table 4, confirming the application of geostatistics to spatial behavior phenomena [60]. DM showed a spatial dependence interval of 13.6 m (Table 4). Thus, there was greater spatial dependence on the distance in the sample grid (10 m × 10 m). The GM showed a weak spatial dependence, adjusted by a linear model with a range of 47.34 m.

### 3.2. Kriging Mapping

Soil nutrient availability for plants depends on adequate pH values (see, e.g., Neina et al. [61]). Figure 2 shows the spatial variability maps, obtained by Kriging interpolation, for pH values at soil depths of 0–0.2 m, 0.2–0.3 m, and 0.3–0.4 m, as well as DM and GM (in ton/ha^−1^), in the seven paddocks.

The lowest soil pH values (4.4) were observed in paddocks six and seven, at depths of 0.2–0.3 and 0.3–0.4 m. Paddock one presented pH values between 5 and 5.6 for all evaluated depths. Paddock four also gave similar results to paddock one, albeit only in the soil layer 0–0.2 m, while the deeper layers (0.2–0.3 m and 0.3–0.4 m) presented a pH reduction, with values ranging from 4.55 to 5.15. Paddock 2 showed a more acidic soil pH (4.85) in the first layer (0–0.2 m), and higher values in the deeper layers (>0.2 m), where the pH ranged from 5 to 5.57.

The spatial distribution of soil pH (Figure 2a–c) showed lower soil acidity areas, with pH values from 5 to 5.63, grouped in the western side of the map. In addition, areas with soil pH values ranging from 4.4 to 4.85 were mostly on the east side of the map.

The maps in Figure 2b,c represent the 0.2–0.3 m and 0.3–0.4 m soil layers, respectively, which presented similar patterns between layers and paddocks. Analyzing the 0.2–0.3 m soil depth (Figure 2b), the third paddock presented a circular area, similar to the 0–0.20 m layer, in the light and dark orange color scales, representing soil pH between 4.55 and 4.7, respectively. At the deepest layer of 0.3–0.4 m (Figure 2c), the pH ranged from 4.85 to 5.15.

Figure 2d shows the spatial variation of DM. Colors in green are the highest amounts of pasture DM, between 9 and 12 ton/ha^−1^, which were observed at the map edges (north and south), located at the paddocks entrances. Paddocks 1 and 6 (Figure 2d) presented more significant regions with light green tones (9–10.5 ton/ha^−1^), surrounded by dark green areas (representing 12 ton/ha^−1^). Lower yields were observed in paddock 3 (Figure 2d), with DM yield between 6 and 4.5 ton/ha^−1^. Under more critical conditions, paddock 7 showed lower yields, compared to the others (3 ton/ha^−1^), in the central area. Paddock 6 (Figure 2d) presented pasture regions with green spots with a better production (10 ton/ha^−1^), compared to the lower (south) paddock part, where the yield decreased from 7.5 to 4.5 ton/ha^−1^. Paddock 5 (Figure 2d) showed greater spatial variability in DM yield, ranging from 4.5 to 9 ton/ha^−1^.

The GM map (Figure 2e) showed that paddocks 1 and 4 had higher forage yields, with values between 21 and 33 ton/ha^−1^, represented by shades of green. Meanwhile, paddocks 3 and 7 presented lower forage yields, with a production of 12 to 21 ton/ha^−1^, represented by the colors in shades of beige and orange. A similar forage distribution spatial pattern can be observed when comparing the DM (Figure 2e) and GM (Figure 2f) maps.

Comparing the soil pH maps (Figure 2a–c) and DM map (Figure 2d), it can be observed that paddocks 3 and 7 showed lower DM values (Figure 2b). Soil pH showed values of 4.85 and 4.4 in all layers (Figure 2a–c). In the western region of the DM map (Figure 2d), represented by paddock 1, the yield was higher than in the other paddocks. In addition, the soil pH map at depths of 0–0.2 m, 0.2–0.3 m, and 0.3–0.4 m (Figure 2a–c) presented better results, with values above 5.15.

The correlation graphs between the pH variables at different depths (0–0.2 m, 0.2–0.3 m, 0.3–0.4 m), and the DM and GM of the forage are shown in Figure 2f–k. The best correlations between DM and pH were found in the soil layers of 0.2–0.3 m and 0.3–0.4 m (Figure 2g,h). The correlation presented in Figure 2f between forage DM and pH at a depth of 0.2–0.3 m showed the highest R^2^ value, equal to 0.31. GM and pH showed a lower relationship, with R^2^ = 0.014, 0.36, 0.05 at depths of 0–0.2 m, 0.2–0.3 m, and 0.3–0.4 m, respectively.

### 3.3. Remote Sensing Mapping

The DM and GM maps of forage were used to analyze the possible relationship between them and the NDVI and NDWI spatial patterns. Evidence of spectral variations can contribute to identification of areas for different management forms. Thus, to identify forage stresses and provide a means of investigating the spatial variability in DM production, maps of NDVI and NDWI were constructed. The pasture regions subjected to spatial data analysis by the application of vegetative indices are shown in Figure 3.

The vegetative index, DM, and GM map combinations indicated considerable spatial variations in the NDVI values. The sampling sites in the vegetation area with the lowest forage mass, represented by red colors (Figure 3a–d), were the same paddocks with the lowest soil pH values (4.55 and 4.4) at all soil depths (Figure 3a–c); therefore, a more acidic soil was seen to be less favorable for forage development.

The vegetative index relationships (Figure 3a,b) indicated a unit reduction in NDVI and NDWI values with decreasing DM (Figure 3c) and GM (Figure 3d) of 5.42 ton/ha^−1^ and 0.87 ton/ha^−1^, respectively. This indicates that DM production contributed to the variability in the vegetation index values.

The regression graphs (Figure 3e–h) between vegetation indices and forage mass variables showed a more significant correlation of NDVI with DM and GM (R^2^ = 0.30, 0.37, respectively). Meanwhile, the NDWI presented an R^2^ of 0.18 for DM and 0.27 for GM.

## 4. Discussion

### 4.1. Geostatistical Analysis

The geostatistical parameters of pH, DM, and GM (Table 3 and Table 4) all showed heterogeneous spatial distributions. The GM was adjusted to the linear model, which has no sill variance and indicated the significant heterogeneity of this variable. The spherical model was fitted to the DM data (Table 4). This model has also been used, by Cavallini et al. [62], to evaluate DM in a Brachiaria pasture. Semivariogram adjustment to the spherical model for soil pH in the 0–0.20 m layer (Table 4) was similar to that in the studies developed by Grego et al. [63], Bernardi et al. [64], and Bernardi et al. [65] when evaluating the spatial variability of soil properties under pastures. Souza et al. [59] and Trangmar et al. [66] have explained that the spherical model better describes the semivariogram behavior when applied to soil attributes commonly used in soil chemical attribute studies.

A strong degree of spatial dependence, as classified by Cambardella et al. [53], was observed for all variables (see Table 3 and Table 4); therefore, the spatial distribution was not random [67]. The ranges (a) for pH (Table 3) and DM (Table 4) indicated that it was higher for soil pH. According to Ferraz et al. [36], the range can be understood as the limiting distance at which a sampled point correlates with other points nearby. Therefore, range values can be used to define sample densities for future studies [42].

### 4.2. Kriging Mapping

In paddocks 5, 6, and 7 (Figure 2a), the soil pH values varied between 4.4 and 4.8, indicating acidic soil in the 0–0.2 m layer. The ideal pH for crops ranges between 5.3 and 6.6; below this range, lime application is required [68]. The research of Grego et al. [63], in which the spatial distribution of pasture soil attributes was analyzed, showed that the soil layer of 0.2 m typically presents pH values in the range of 4.0–5.6. In the deeper soil layers of 0.2–0.3 m and 0.3–0.4 m (Figure 2b,c), a pH value of the soil ranging between 4.4 and 4.7 was observed in paddocks 4–7, characterizing the soil in these layers as being more acidic. In acidic soils, plants can be affected by aluminum (Al) toxicity and reduced nutrient availability, leading to undesirable reductions in pasture growth [69,70].

In a similar study, Oliveira et al. [68] observed pH values between 4.0 and 5.0 and a relationship with dry matter accumulation. Serrano et al. [71] mapped soil attributes in a pasture, and indicated a pH between 5.49 and 8.02 under pasture cultivation. Some authors have explained these variations as being attributable to extrinsic factors, such as fertilization and correction reflected in soil chemical characteristics. Pasture soils are sensitive to additional nitrogen (N) deposition, resulting in a loss of nitrate and depletion of base cations, leading to elevated soil acidity [72,73]. As there were added chemical fertilizers in the studied pasture, the soil pH variation (Figure 2a–c) may be related to the residual nitrogen fertilization effect, which was carried out periodically in the pasture, influencing the formation of areas with high concentrations of fertilizer, inducing changes in soil pH and DM.

Additionally, the pH changes (Figure 2a–c) may be related to the trampling effect caused by the movement of animals inside the paddocks. According to Campbell et al. [74], a buffalo moves up to 10 km daily, driven by the abundance and distribution of forage. In addition, they may habitually revisit grazed areas, resulting in extensive use. Villalobos-Barquero et al. [75] explained that the animal’s naturally excessive weight causes compaction in the soil layers, derived from the impacts of hooves. Compaction is characterized by increased soil mechanical resistance to penetration (RP), making it difficult for nutrients to percolate into the soil and causing acidity [76]. In addition, increased RP influences mass production in forage [63].

The spatial vegetation heterogeneity shown on the map (Figure 2d), therefore, may have been influenced by factors such as fertilization, trampling, and selective grazing [77]. Buffalo management in limited spaces can cause some negative characteristics in the breeding environment, as a result of their hooves (e.g., running over, channeling, and soil compaction) [74]. Pastures are typically subject to stressful conditions, resulting in altered forage growth patterns [78]. Additionally, soil fertility factors and moisture gradients also result in spatial vegetation variations [79,80].

Mombaça grass has a high productive response to nitrogen fertilization [10]. Nitrogen fertilizer application may cause nitrogen supply accumulation in the soil in small pasture areas, altering the DM accumulation rate. The discontinuous productivity of forage within a paddock leads to an irregular supply of forage, interfering with buffalo milk production. According to Silva et al. [81], animal feed is associated with herd production and adequate feeding of buffaloes, and may interfere with production requirements by decreasing daily milk production. Lima et al. [82] stated that keeping lactating buffalo herds in areas with limited access to quality forage could affect their milk production and quality.

Precision livestock systems are being implemented in modern agricultural systems, and have been gradually adopted in extensive agriculture [83]. In precision livestock, fertilizer application at a variable rate is carried out systematically, as a uniform application can result in abrupt variations in fertilization in certain areas [84].

The mapping of pasture and soil variables in rotated paddocks represents an initial advance for biomass variation identification within the paddock, in addition to allowing for the identification of areas with high soil acidity. From this mapping, inputs and variable rate applications can become a reality. In addition, paddock stocking and rotation strategies can be traced by following the pH and DM parameters. According to Bernardi et al. [64], DM production maps can be used to avoid over- or under-grazing, allowing for the estimation of stocking and production rates within the area.

The most significant similarity between the spatial standards and the variables was between pH 5.63 and DM 9–12 ton/ha^−1^. According to Bailey et al. [85], soils with a pH value of 6.0 in cultivated pastures contribute to expressive biomass development.

Soil pH is the parameter most likely to affect vegetative growth [86]. When measuring the proportion of DM variance explained by soil pH, it was found that the coefficient of determination (R^2^) showed negative values for all soil layers. In addition, pH correlations with dry mass indicated that the considered model does not explain the variation in the DM variable. In this case, pasture nutrition responds not only to soil acidity, but also to other factors such as nutritional balance and physical characteristics. Research carried out by You et al. [73] showed that the forage mass response ratio was not significantly correlated with changes in soil pH, indicating that an increase in forage mass cannot be directly related to soil pH.

The geostatistical mappings in this study can be used to adjust the animal stocking rate, according to the forage restrictions in each paddock. The advantages of this methodology include identifying areas for the variable management of paddocks [65].

### 4.3. Remote Sensing Mapping

According to the coefficient of determination (R^2^), the NDWI variation showed a low correlation with the GM and DM data, indicating that the NDWI explains 18% of the DM and 27% of the GM variation, respectively. This index was used in the study of Tong et al. [87], where it showed a good correlation with superior forage biomass characteristics. The low NDWI ratio in this study may be related to the formation of pastures in the soil. Vegetative indices provide valuable spatial and temporal information on a large scale regarding yield-limiting factors and crop response, but some indices can be influenced by soil moisture conditions and rangeland uniformity [88].

It was observed from the regression graphs (Figure 3e,g) that the NDVI performed better compared with the NDWI (Figure 3a,b, respectively). The biophysical relationship between spectral refraction of vegetation indices and DM is not direct. DM is a qualitative parameter; therefore, it is related to several factors [85]. The changes in DM cannot be directly observed through changes in reflectance, as it influences the GM, which is a quantitative parameter that involves the amount of fresh forage mass with water in the cell. Vegetation index maps are efficient for investigating the spatial variability in pasture GM production. Thus, remote sensing data can be used to predict GM yield, instead of having to cut forage samples and weigh the mass at each sampling point.

The advent of hyperspectral sensors in satellite platforms has raised new expectations for improving pasture productivity estimates [24]. The enhanced spectral wave range in sensors could lead to innovations in the sector, providing variations not observable by multi-spectral sensors. Campbell et al. [74] used the NDVI to provide a measure of vegetative vigor in buffalo activity spaces, as a parameter for determining the quantity and quality of forage available. They demonstrated that it could be used to measure temporal variations in forage availability reliably. A study by Tong et al. [87] showed the high potential of combining hyperspectral vegetation indices for the estimation of pasture biomass.

Wang et al. [28] estimated the pasture green mass using optical remote sensing data. Wagle et al. (2020) used a vegetation index derived from satellite imagery at different spatial and temporal resolutions in order to examine pasture heterogeneity within and between paddocks, and verified its consistency for the detection of vegetation phenology. Webb et al. [86] observed that crops stabilized above a soil pH of approximately 6, and there was little discernible difference in NDVI. This result is relevant for identifying management zones with pH > 6.

## 5. Conclusions

A precision agriculture approach based on geostatistics and vegetation index analysis tools allowed for the prediction of spatial variability for dry matter (DM) in paddocks for buffaloes cultivated with *Panicum maximum* (Jacq.) cv. Mombaça. Kriging maps showed that areas with pH below 4 present lower forage production. Vegetation indices made it possible to identify pasture areas with lower vigor and, consequently, lower forage production. Therefore, this technology could identify field areas with soil pH below 4, potentially reducing time and cost. These are significant resources for agricultural producers, allowing for timely remediation of areas with low pH content. By applying the approaches proposed in this article, it was possible to spatially characterize the forage productivity within a pasture system in which buffalos are rotated.

The quantification of GM and DM proved efficient in evidencing the spatial variation of forage production. In addition, they can be used as predictor variables to assess the spatial variability of the NDVI. These results demonstrate the high potential of multi-spectral data in terms of estimating the vigor of Mombaça grass pastures at the canopy level.

## Figures and Tables

**Figure 1 animals-12-02374-f001:**
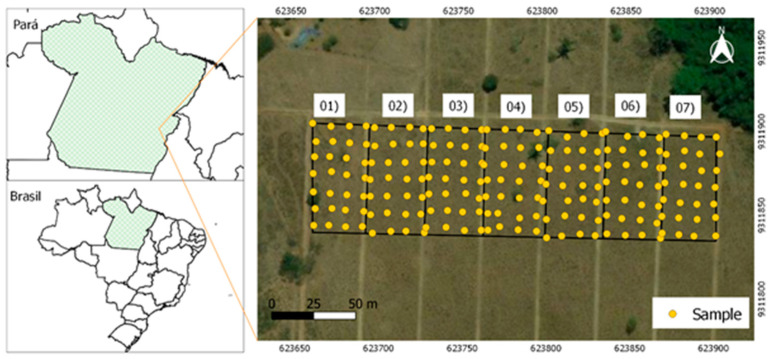
Location of the study area and experimental divisions.

**Figure 2 animals-12-02374-f002:**
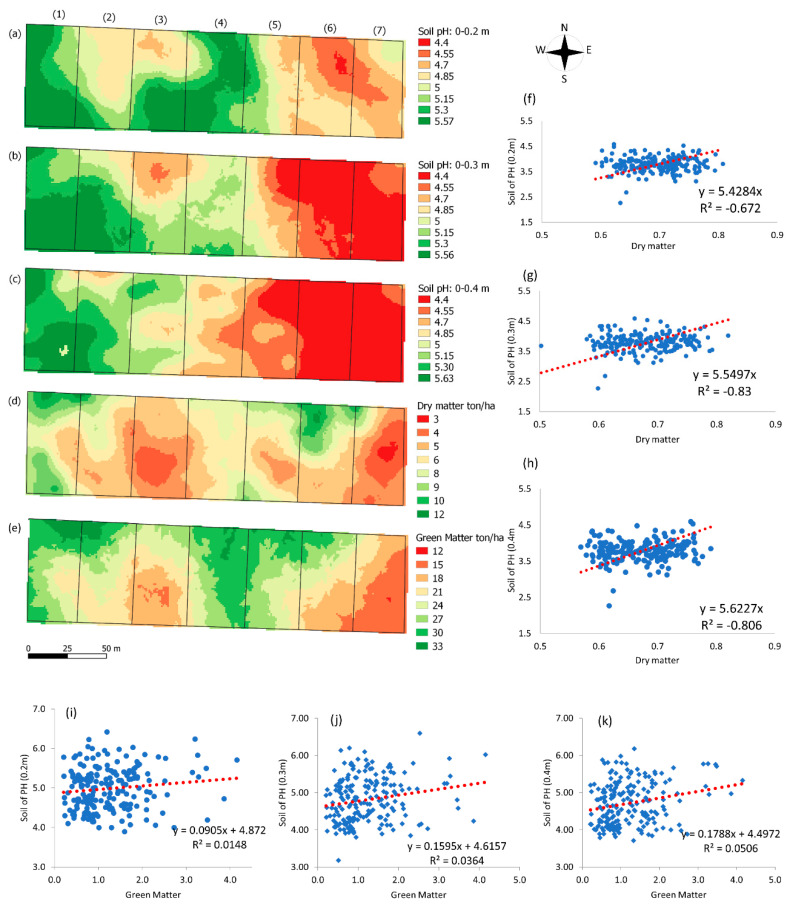
Kriging maps of spatial pH variation at soil depths of 0.2 m (**a**), 0.3 m (**b**), and 0.4 m (**c**); DM (dry matter ton/ha^−1^) map (**d**); GM (green matter ton/ha^−1^) map (**e**); and regression graphs between DM, GM, and soil pH at a depth of 0.2 m (**f**,**i**), 0.3 m (**g**,**j**), or 0.4 m (**h**,**k**).

**Figure 3 animals-12-02374-f003:**
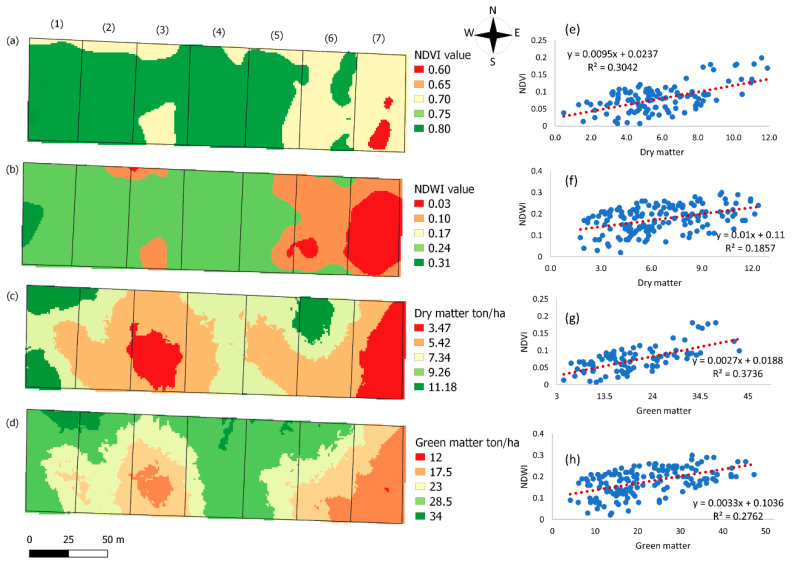
NDVI map (**a**); NDWI map (**b**); DM Kriging map (dry matter, ton/ha^−1^) (**c**); GM Kriging map (green matter, ton/ha^−1^) (**d**). Regression graphs between DM (dry matter) and NDVI (**e**); DM (dry matter) and NDWI (**f**); GM (green matter) and NDVI (**g**); and GM (green matter) and NDWI (**h**).

**Table 1 animals-12-02374-t001:** Soil granulometric analysis at depths of 0.0–0.2 and 0.2–0.4 m.

Depth (m)	Total Sand (g/kg)	Total Clay (g/kg)	Silt (g/kg)
0.0–0.2	676.85	173.30	149.84
0.2–0.4	656.05	150.50	135.82

**Table 2 animals-12-02374-t002:** Sentinel 2B sensor’s spectral and spatial resolutions characteristics.

#	Band Description	Central Wavelength (nm)	Bandwidth (nm)	Spatial Resolution (m)
1	Aerosols	442.2	21	60
2	Blue	492.1	66	10
3	Green	559.0	36	10
4	Red	664.9	31	10
5	Red-edge 1	703.8	16	20
6	Red-edge 2	739.1	15	20
7	Red-edge 3	779.7	20	20
8	Near infra-red	832.9	106	10
8a	Red-edge 4	864.0	22	20
9	Water vapor	943.2	21	60
10	Cirrus	1376.9	30	60
11	SWIR 1	1610.4	94	20
12	SWIR 2	2185.7	185	20

**Table 3 animals-12-02374-t003:** Semivariogram models and parameters adjusted to values obtained from soil pH at soil depths of 0–0.2 m, 0.2–0.3 m, and 0.3–0.4 m.

Depths (m)	Model	a	a′	C_0_ + C_1_	C_0_	R^2^	RSS	DSD
0–0.2	Sph	83.90	83.90	0.404	0.194	0.92	1.62	Strong
0.2–0.3	Gaus	110	256.91	0.705	0.220	0.98	1.93	Strong
0.3–0.4	Gaus	235.69	298	0.851	0.149	0.98	1.60	Strong

Models: Sph, spherical model; Gaus, Gaussian model; a, range (m); a′, practical range; (C_0_ + C_1_), sill variance; C_0_, nugget effect, R^2^, coefficient of determination; RSS, residual sum of squares; DSD, degree of spatial dependence.

**Table 4 animals-12-02374-t004:** Model and semivariogram parameters, adjusted to obtain values from dry and green matter (ton/ha^−1^).

	Model	a	a′	C_0_ + C_1_	C_0_	R^2^	RSS	DSD
DM	Sph	13.6	13	26,240,000	530,000	0.926	1.22	Strong
GM	Lin	47.34	47.34	130.288	97.377	0.4	1.23	Weak

Models: Sph, spherical model; Lin, linear model; a, range (m); a’, practical range (m); (C_0_ + C_1_), sill variance; C_0_, nugget effect, R^2^, coefficient of determination; RSS, residual sum of squares; DSD, degree of spatial dependence.

## Data Availability

Not applicable.

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
