# Peer review of "Mapping Soil and Pasture Attributes for Buffalo Management through Remote Sensing and Geostatistics in Amazon Biome"

_animals, 2022, doi:10.3390/ani12182374_

Round 1

Reviewer 1 Report

Technical comments:

1.      In Figures 2 (f), (g) and (h), the linear fitted model is missing the intercept value.

2.      In Figure 2, the R-squared (coefficient of determination) is written as a negative number, which is wrong. Please check your calculations. The value of R-squared ranges from 0 to 1 and cannot be negative.

3.      Please elaborate on the reasoning for using soil pH as a measure of soil forage production. Why not use other parameters like soil water content, organic matter, nutrient distribution etc.?

4.      In Figures 2 (f), (g), (h), show a few outlier cases that are skewing the slope. For the majority of the data points, DM/GM and pH appears to have poor correlation, please explain the criteria used for removing any outliers, if not, why?

5.      In the conclusion you mentioned that “The kriging maps showed that areas with pH below 4 have lower forage production“. However, the correlation between soil pH and Dry matter (DM)/Green matter (GM) is poor (R-squared is low) which suggests that based on a linear model, pH has minimal effect on DM and GM, please explain how you arrived at the conclusion mentioned in the article.

6.      Following on the previous point, it is mentioned in Line 415-416 that “Research carried out by You et al. [74] showed that forage mass response ratio was not significantly correlated with changes in soil pH. This cause indicates that the increase in forage mass cannot be directly related to soil pH.” However, your conclusions do not align with this, please elaborate.

7.      In Figure 3 the linear fitted model is missing the intercept value.

8.      In Figure 3, the R-squared (coefficient of determination) is written as a negative number, which is wrong. Please check your calculations. The value of R-squared ranges from 0 to 1 and cannot be negative.

9.      NDVI is a vegetative index and therefore appears to be affected by DM and GM as presented in Figure 3. Please explain how does pH measurement integrate with NDVI measurement.

Other comments:

1.      Overall, the English needs significant improvement throughout the article. Many sentences do not make sense and/or are hard to understand. I strongly recommend that you find a native English speaker to thoroughly check your manuscript.

2.      Please provide the full forms for the abbreviations before using them. Following is a list of abbreviations that should be defined before they are used:

a.      Line 37 uses NDVI and NDWI without prior definition.

b.      UTM in line 107

c.      AW in line 108

d.      KCl in line 132

3.      Please use subscript or superscript to correctly represent units like ton/ha-1 throughout the manuscript.

4.      Please improve the quality of all Figures, particularly the calibration graphs in Figures 2 and 3.

Author Response

Response rev1

Reviewer 2 Report

Reviewed manuscript "Mapping soil and pasture attributes of Buffalo's management by remote sensing and geostatistics in Amazon Biome" (animals-1842369) contains the results of very interesting research work of scientific and practical significance.

Precision agriculture technologies to assess soil pH and the biomass spatial variability were studied. The experiment was conducted in an area cultivated with Panicum maximum (Jacq.) Cv. Mombaça in a rotational grazing system for dairy buffaloes in the eastern Amazon.

The experiment was planned properly and carried out on sufficiently numerical material.

Statistical analysis of the obtained results is correct.

Tables and figures presented the results and statistical data were constructed properly.

The discussion was carried out properly and the literature used in this part of the manuscript was chosen accordingly.

In summary - the manuscript should be published in Animals.

Author Response

Thank you.

Reviewer 3 Report

 General comments

This study aimed to apply precision agriculture in Amazon biome regarding Panicum maximum (Jacq.) cv. Mombaça biomass and soil pH.

This study was very well designed and have scientific soundness. Nevertheless, some issues need to be clarified mainly at results and discussion level. 1) the regression analysis should be identified (presumably Pearson correlations); 2) The P-values are lack in the results; there are a confusion in R2 coefficients (there are presented as a negative, but neither there are close to zero, nor they represent multiple linear regressions; also see the line inclination); 3) In discussion, the interpretation of these regressions is sometimes confusing. This requires a minor/moderate revision.   

Specific comments

L214: Please remove the point

L269 (Fig.2): The coefficient regression for dry matter is negative (also in fig. 3g). Please correct. Also correct the y-axis, if possible.

L308: R2=0.014. also correct L332-333.

L335: You mean a “more intensive use”?

L376: This excessive weight can be also caused by the animal’s trampling effect?

L390. Why this can occur only in “small pasture areas”? Only due the “irregular supply of forage”? This variation is negatively related with precision livestock (L397-399)? I suggest to clarify this important aspect.

L412: According to your graphs in Fig 2. The DM increase as the pH increase. Note that all values are below pH=6.

L413: According to your data, in Fig. 2f, g and h, the pH variation justify 67% (R2=0.67), 83% and 81%, respectively, of the dry mater increment, and indicating that the lower pH observed in this study is a major factor of nutrients´ availability.  Your statement is only right for green matter. The P-values were omitted in your data and need be added (e.g., we don’t know if small R2 value are significant or not).

L424: “…GM in 5.5%, respectively.”

Author Response

Response rev3
